# In-situ cross-linking strategy for efficient and operationally stable methylammoniun lead iodide solar cells

Xiaodong Li [1], Wenxiao Zhang[1,2], Ying-Chiao Wang [1], Wenjun Zhang[1,2], Hai-Qiao Wang[1,2] & Junfeng Fang[1,2]

Long-term operational stability is the foremost issue delaying the commercialization of perovskite solar cells (PSCs). Here we demonstrate an in-situ cross-linking strategy for operationally stable inverted $MAPbI_3$ PSCs through the incorporation of a cross-linkable organic small molecule additive trimethylolpropane triacrylate (TMTA) into perovskite films. TMTA can chemically anchor to grain boundaries and then in-situ cross-link to a robust continuous network polymer after thermal treatment, thus enhancing the thermal, water-resisting and light-resisting properties of organic/perovskite films. As a result, the cross-linked PSCs exhibit 590-fold improvement in operational stability, retaining nearly 80% of their initial efficiency after continuous power output for 400 h at maximum power point under full-sun AM 1.5 G illumination of Xenon lamp without any UV-filter. In addition, under moisture or thermal (85 °C) conditions, cross-linked TMTA-based PSCs also show excellent stability with over 90% of their initial or post burn-in efficiency after aging for over 1000 h.

[1] Ningbo Institute of Materials Technology and Engineering, Chinese Academy of Sciences, Ningbo 315201, China. [2] University of Chinese Academy of Sciences, Beijing 100049, China. Correspondence and requests for materials should be addressed to J.F. (email: fangjf@nimte.ac.cn)

Organic–inorganic metal halide perovskite solar cells (PSCs) are regarded as one of the most promising candidates in photovoltaic field due to their low-cost and high efficiency[1–6]. The device stability is a major bottleneck limiting the large scale development of PSCs[7,8] since the certified power conversion efficiencies (PCEs) have exceeded 20%[9–13]. The air (moisture) and thermal stability of PSCs have been significantly improved in past few years[13–17]. However, it is still far behind the requirement of commercialization as the most crucial issue of operational stability, namely the continuous power output of PSCs when subjected to realistic working conditions with light illumination and external load remains a challenge. Recently, through suppressing the degradation at charge transport layer (CTL), great improvement has been achieved in operationally stable PSCs by replacing the organic CTL with inorganic materials such as CuSCN or chlorine-caped $TiO_2$[18,19], which is the first but encouraging step toward operational stability. Studies have demonstrated that the stability issues of PSCs appear not only in CTL, but also in perovskite layer[20]. Up to now, the limited reports on long-term operational stability mainly aim at the CTL-induced degradation[18,19] and rare reports are conducted from the point of perovskite layer. To further improve the operational stability, study on perovskite layer is necessary since the degradation of perovskite layer will impose limitation on the stability of PSCs, once the degradation caused by CTL is suppressed.

Solution-processed perovskite films usually have large grain boundaries (GBs), which are energetically unstable and easy to be attacked[11,21]. To improve the stability of perovskite films, one effective strategy is to cap these GBs with suitable protective materials[22,23]. Among them, small molecule additives that own weak interaction with GBs have been widely used, such as pyridine[24,25], ammonium chloride[26], alkylphosphonic acid ω-ammonium chlorides[23], tertiary or quaternary hydrophobic alkyl ammonium cations[11,27] and phenylalkylamine[28,29]. In addition, linear polymers of polyethyleneimine (PEI) and poly(4-vinyl-pyridine) (PVP) are also reported as additives for use in PSCs[30,31], although it may lead to the precipitation in perovskite precursor solution due to their strong interaction with $PbI_2$. The reported additives can passivate the defects and form a water-resisting layer at GBs to block the moisture penetration. As a result, the device efficiency and air stability is significantly improved. However, these reports mainly aim at the water-resisting property of additives to improve the moisture stability of PSCs; while their operational stability at maximum power point (MPP) is presented merely in time scale of several hours[11,28,30] or even less (for example, 200 s)[24–27,31], which is far behind the requirement of commercial application. For operational stability, more factors, apart from moisture should be considered, including thermal, electric, light and so on[8,13,16]. Given the fact that cross-linked polymers usually possess excellent mechanical, thermal, dielectric and light-resisting properties[32–34], it is envisaged that the cross-linking of organic additives should be a feasible method to improve the related properties of perovskite films and thus enhance the operational stability of PSCs.

Here, we develop an in-situ cross-linking strategy of organic additives to improve the operational stability of perovskite films. In our strategy, the cross-linkable monomer trimethylolpropane triacrylate (TMTA, Fig. 1a) is mixed into perovskite precursor solution and deposited on substrate to obtain perovskite films with TMTA, which can be further cross-linked through thermal treatment (Fig. 1b). As a result, highest efficiency approaching 20% is obtained in PSCs with TMTA. More importantly, the devices exhibit 590-fold improvement in operational stability relative to control devices, retaining nearly 80% (81.6% from J–V curve) of initial efficiency after continuous power output at MPP for 400 h under full-sun AM 1.5 G illumination (100 mW cm$^{-2}$).

Apart from operational stability, the air (relative humidity of 45–60%) and thermal (85 °C) stability are also greatly improved, retaining over 90% of the initial or post burn-in efficiency after aging for over 1000 h. Different to most of the previous reports using ultraviolet-free white LED lamp[18,35,36] or Xenon lamp with UV-filter[19], our operational stability test is conducted under standard Xenon lamp (Newport Oriel Sol3A solar simulator) without any filter and a constant load (0.84 V) is continuously applied on the PSCs to simulate the realistic working conditions. In this work, we use methylammoniun lead iodide perovskite material ($MAPbI_3$) given its simple composition and easy fabrication. Compared with formamidinium lead iodide ($FAPbI_3$), the $MAPbI_3$ device performance will not be greatly affected by the tetragonal-cubic phase transition due to their both black photoactive phase[37,38]. Despite the above-mentioned advantages, no studies are reported about long-term operational stability in $MAPbI_3$ until now. In addition, we adopt inverted device architecture in this work to avoid the use of high temperature sintering $TiO_2$ in norm devices.

## Results

**In-situ cross-linking strategy**. There are three unique advantages of TMTA: first, TMTA is a sticky liquid at room temperature (Supplementary Fig. 1a). During the crystallization of solid perovskite, the liquid TMTA will be automatically expelled to GBs without the interruption of crystal growth. Second, the carbonyl groups in TMTA allow weak interaction with $PbI_2$ (Fig. 1c), making TMTA chemically anchor to GBs and passivate the defects[39], thus leading to improved device efficiency of over 20%. Third, the three alkenyl groups in TMTA allow for the in-situ cross-linking polymerization (Fig. 1c) to process at GBs under moderate thermal conditions (140 °C), enhancing the thermal, water-resisting and light-resisting properties of perovskite film and thus improving operational stability of PSCs.

Fourier transform infrared spectroscopy (FTIR) is collected in $MAPbI_3$–TMTA films before and after thermal treatment to verify the in-situ cross-linking of TMTA (Fig. 2a). Pure TMTA exhibit characteristic peaks of $–CH_3$ (2970 cm$^{-1}$), $C=O$ (1734 cm$^{-1}$) and $CH_2=CH$ groups. Among them, the vibration peaks of $CH_2 = CH$ groups are in-depth studied, including the C–H stretching vibration of $v_{=CH}$ and $v_{=CH_2}$, C–C stretching vibration of $v_{C=C}$ and the C–H bending vibration of $\gamma_{=CH_2}$. In $MAPbI_3$–TMTA films, the vibrations peaks ascribed to $CH_2 = CH$ all disappear after cross-linking at 140 °C (Fig. 2a and Supplementary Fig. 2a, 2b), including $v_{=CH}$ (3110 cm$^{-1}$), $v_{=CH_2}$ (3043 cm$^{-1}$) and $\gamma_{=CH_2}$ (903 cm$^{-1}$), while the peaks of $–CH_3$ and $C=O$ still exist. This result indicates that TMTA in perovskite films indeed cross-link with each other via $CH_2=CH$ groups when annealed at 140 °C. Note that N–H vibration in $MAPbI_3$ also appears around 1640 cm$^{-1}$ (Supplementary Fig. 2c) which overlaps with the C–C vibration of $CH_2 = CH$ in TMTA ($v_{C=C}$ 1640 cm$^{-1}$)[40]. The peak at 1640 cm$^{-1}$ is greatly weakened in $MAPbI_3$–TMTA films after cross-linking due to the polymerization of $CH_2 = CH$ groups and the small peak around 1640 cm$^{-1}$ should be ascribed to the N–H vibration in $MAPbI_3$ (magnified FTIR in Supplementary Fig. 2d). In addition, the liquid TMTA will become a solid after annealing at 140 °C, strongly confirming its cross-linking under thermal conditions (Supplementary Fig. 1b).

**Perovskite films characterization**. In order to realize low-temperature processed PSCs, inverted device architecture is adopted on the base of our previous works, with poly[3-(4-methylamine carboxylbutyl)thiophene] (P3CT-N)[41] and [6,6]-phenyl C$_{61}$ butyric acid methyl ester (PCBM) as hole and electron transport layer, respectively (Fig. 2b)[42]. $MAPbI_3$ precursor

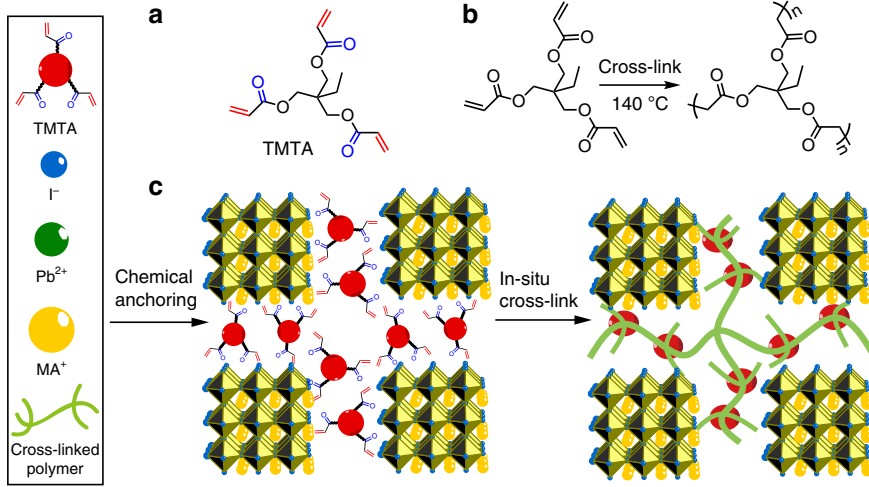

**Fig. 1** Schematic illustrations of in-situ cross-linked organic/perovskite films. **a** Chemical structure of TMTA with marked carbonyl (blue) and alkenyl (red) groups. **b** Cross-linking polymerization of TMTA under thermal conditions. **c** Working mechanism of TMTA in PSCs: TMTA chemically anchors to the grain boundaries of MAPbI$_3$ and then in-situ cross-links to a continuous network polymer

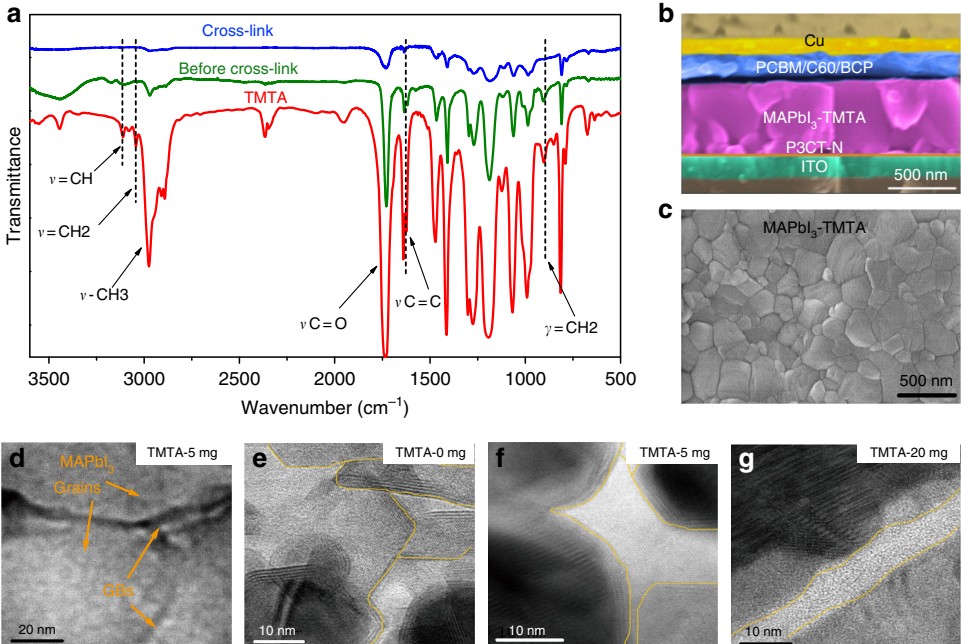

**Fig. 2** FTIR and morphology characterization. **a** FTIR spectra of MAPbI$_3$–TMTA films before (olive), after (blue) cross-linking and pure TMTA (red). The dash lines indicate the characteristic vibration peaks of TMTA. **b** Devices structure and cross-section SEM images of inverted PSCs used in our work. **c** Top-view SEM images of MAPbI$_3$–TMTA films. **d** Low-magnification TEM image of MAPbI$_3$–TMTA films. TMTA-5 mg indicates that the concentration of TMTA in MAPbI$_3$ precursor solution is 5 mg mL$^{-1}$. **e-g** High resolution TEM of perovskite films: (**e**) MAPbI$_3$; (f) MAPbI$_3$–TMTA (5 mg mL$^{-1}$), and (**g**) MAPbI$_3$–TMTA (20 mg mL$^{-1}$). The yellow lines demarcate the grain boundary (GB) regions

solution with TMTA is deposited on ITO/P3CT-N substrate through typical anti-solvent method. Cross-sectional scanning electron microscopy (SEM) images reveal a 450 nm MAPbI$_3$–TMTA layer closely sandwiched between P3CT-N and PCBM interlayer (Fig. 2b). The MAPbI$_3$–TMTA layer is vertically compact without obvious cracks or pinholes, which is beneficial to the vertical carriers transport. We also acquire the top-view SEM images of perovskite film as shown in Fig. 2c. MAPbI$_3$–TMTA exhibits smooth and pinholes free morphology with large grains (average size 236 nm in Supplementary Fig. 3), similar to control MAPbI$_3$ films due to the identical processing method (cross-sectional and top-view SEM of MAPbI$_3$ in

Supplementary Fig. 4). As expected from previous studies[24,25,29], MAPbI$_3$–TMTA shows similar x-ray diffraction (XRD) patterns to MAPbI$_3$ and no obvious shift in diffraction angle is observed (Supplementary Fig. 5), indicating that TMTA additive will not embed into crystal lattice of perovskite and can only exist at the GBs. We further use transmission electron miscroscopy (TEM) to investigate the nanoscale structure of perovskite films. To minimize damage to relatively soft MAPbI$_3$ films[43], the TEM samples are prepared by dropping perovskite solution directly onto carbon-coated TEM grids (details shown in Characterization section). Fig. 2d shows the lower magnification TEM image of cross-linked MAPbI$_3$–TMTA films with obvious MAPbI$_3$ grains

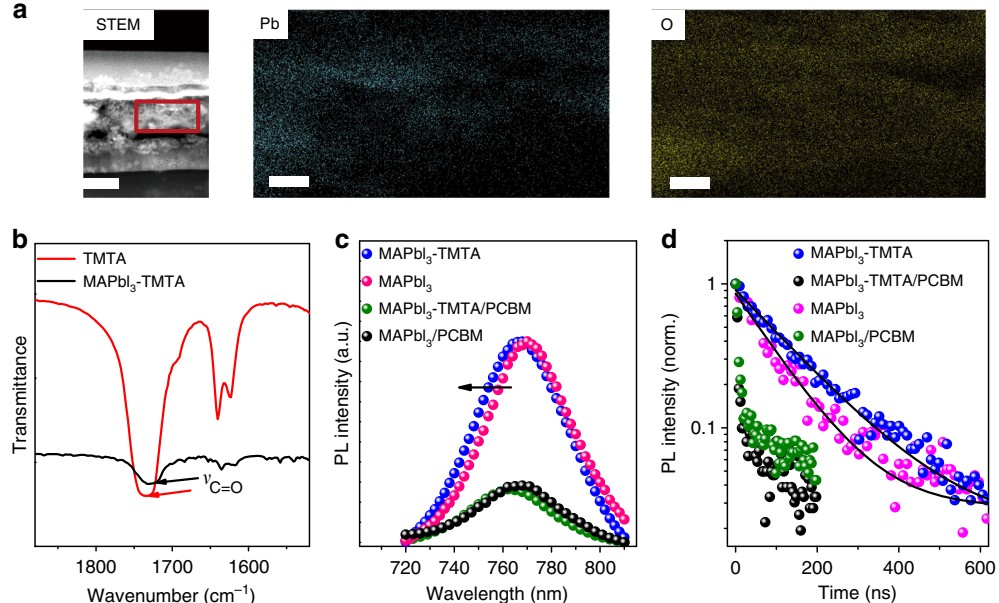

**Fig. 3** TMTA distribution and photoluminescence (PL) characterization. **a** From left to right, scanning TEM (STEM, scale bar, 200 nm) image and EDS mapping of Pb and O (scale bar, 50 nm). The red line demarcates the EDS mapping area. **b** Enlarged FTIR spectra of MAPbI$_3$–TMTA and pure TMTA; the arrows indicate the stretching vibration peak of C=O. **c** Steady-state PL spectra and **d** time-resolved PL decays of MAPbI$_3$, MAPbI$_3$/PCBM, MAPbI$_3$–TMTA and MAPbI$_3$–TMTA/PCBM. The arrow indicates the blue-shifted PL peak in MAPbI$_3$–TMTA film

and grain boundaries (GBs). High resolution TEM images are shown in Fig. 2e–g, focusing on the GB regions. The MAPbI$_3$ films are fully crystalline without any obvious amorphous regions at GBs (Fig. 2e). The MAPbI$_3$–TMTA films (Fig. 2f) clearly show that the amorphous walls, which can be ascribed to cross-linked TMTA, mainly exist among crystalline MAPbI$_3$ grains[43]. And the amorphous TMTA walls become much clearer in MAPbI$_3$–TMTA films with high TMTA concentration (Fig. 2g, more detailed views of HRTEM in Supplementary Fig. 6). This result strongly confirms that cross-linked TMTA indeed exists at GBs in perovskite films[43].

Cross-sectional scanning TEM (STEM) and the corresponding energy dispersive X-ray spectroscopy (EDS) elemental mapping are conducted as shown in Fig. 3a. To in-situ characterize the vertical elemental distribution, STEM samples are prepared by spin-coating perovskite solution on ITO substrate using the same method with device fabrication. Then the sample is further processed using focused ion beam lift-out technique. The mapping area is focused on the perovskite layer, which is demarcated with red lines in Fig. 3a. In STEM-EDS mapping, Pb element represents MAPbI$_3$ phase and O element represents TMTA phase. O element is observed throughout the Pb-rich area, indicating the homogeneous distribution of TMTA. Combining with the results from HRTEM in Fig. 2, it can be concluded that TMTA exists at GBs in the whole perovskite films. To further investigate the existence states of TMTA in perovskite films, we acquire the enlarged FTIR spectra of pure TMTA and MAPbI$_3$–TMTA as shown in Fig. 3b. The stretching vibration of C=O ($v_{C=O}$) in pure TMTA appears at 1734 cm$^{-1}$, while it shifts to 1725 cm$^{-1}$ in MAPbI$_3$–TMTA. The shift to low wavenumber is indicative of the weakened C=O bond as a consequence of the coordination between TMTA and MAPbI$_3$[10]. Such coordination will help TMTA to anchor to the GBs and then passivate the possible defects.

Previous studies have confirmed that the defects at GBs will lead to the red-shift in photoluminescence (PL) spectra[44]. The control MAPbI$_3$ film exhibits a PL peak at 770 nm, while the

MAPbI$_3$–TMTA film exhibits a PL peak at 766 nm (Fig. 3c). The blue-shifted PL peak indicates that TMTA can passivate the defects at GBs due to its coordination with MAPbI$_3$ as confirmed in FTIR. In the presence of electron transport layer (PCBM), the PL of MAPbI$_3$–TMTA is strongly quenched, indicating the efficient electron extraction between bulk perovskite and PCBM. Fig. 3d shows the time-resolved PL (TRPL) of perovskite films on ITO substrate. The longer carriers lifetime indicates better electronic quality in MAPbI$_3$–TMTA (142 ns) than in control MAPbI$_3$ (101 ns). When PCBM is introduced, the lifetime sharply decreases to 5.6 ns in MAPbI$_3$–TMTA/PCBM and to 5.9 ns in MAPbI$_3$/PCBM, indicating the slightly faster electrons transfer at MAPbI$_3$–TMTA/PCBM interface.

**Photovoltaic device performance.** After successfully introducing TMTA into perovskite films, we investigate the photovoltaic performance of PSCs as shown in Fig. 4 and Supplementary Table 1. The PSCs with TMTA before cross-linking exhibit the highest efficiency of 20.22% (Fig. 4a, inset: maximum power output of 19.8%) with an open-circuit voltage ($V_{oc}$) of 1.11 V, short-circuit current density ($J_{sc}$) of 22.8 mA cm$^{-2}$ and fill factor (FF) of 80.2% (Fig. 4a), but their stability is poor (discussed in Long-term stability section below). Therefore, we start to pay more attention to the MAPbI$_3$–TMTA devices after cross-linking in next investigation. The PSCs with TMTA after cross-linking yield a $V_{oc}$ of 1.09 V, $J_{sc}$ of 22.7 mA cm$^{-2}$ and FF of 78.2%, resulting in an overall efficiency of 19.26% (Fig. 4b). The slight decrease in device efficiency after cross-linking may be caused by the insulation nature of cross-linked TMTA. The MAPbI$_3$ devices exhibit a lower efficiency of 19.08% with $V_{oc}$ of 1.09 V, $J_{sc}$ of 22.4 mA cm$^{-2}$ and FF of 78.1% (Fig. 4c). As evident from the J–V curves under forward and reverse scan, the hysteresis is discernable in control MAPbI$_3$ devices, but it is negligible in devices with cross-linked TMTA (reverse efficiency of 19.01%). Furthermore, the PSCs are probed at MPP under AM 1.5 G illumination to determine their stabilized PCE. We obtain a stabilized

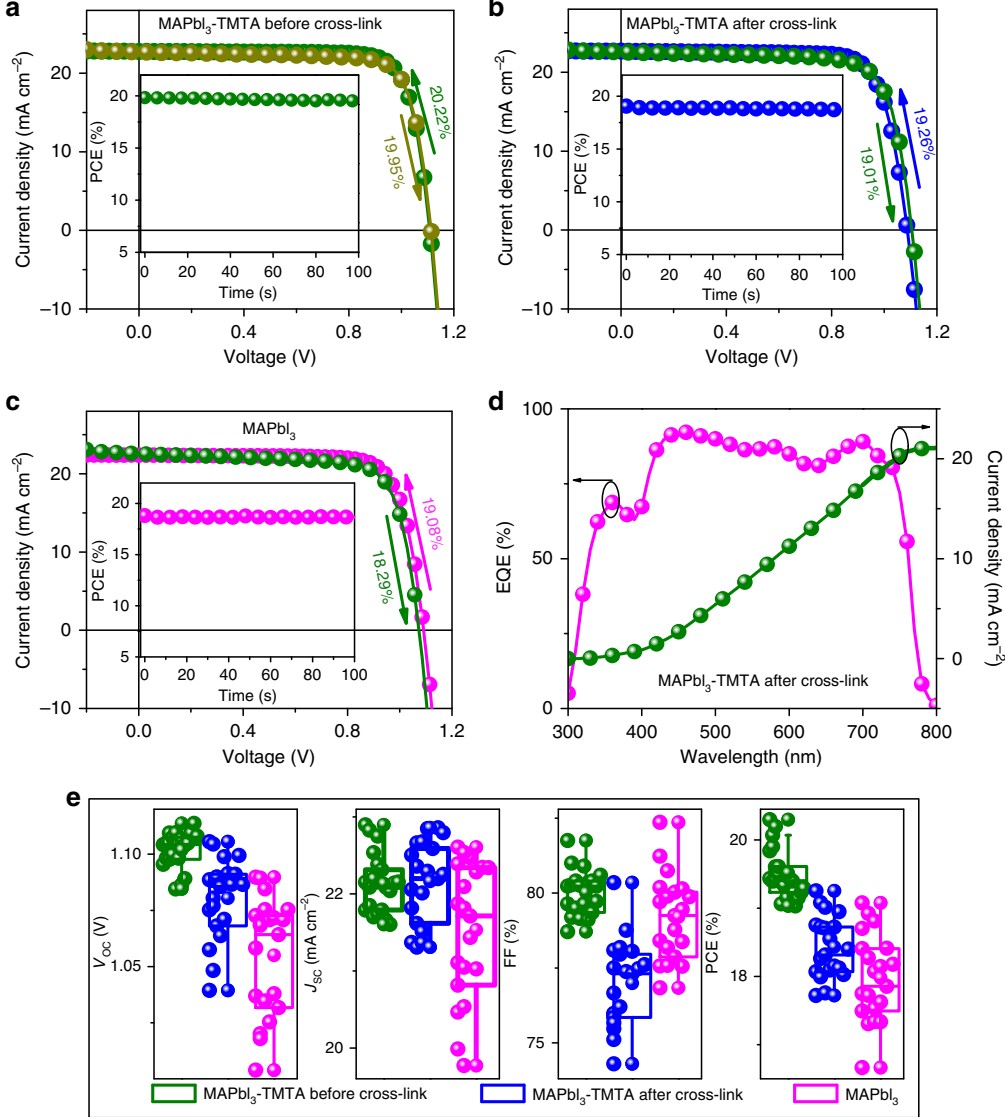

**Fig. 4** Photovoltaic characterization. **a** *J–V* curves of MAPbI₃–TMTA PSCs before cross-linking; **b** *J–V* curves of MAPbI₃–TMTA PSCs after cross-linking; **c** *J–V* curves of control MAPbI₃ PSCs. The arrows indicate the scan direction of *J–V* measurement. Inset: the stabilized efficiencies of PSCs at MPP. **d** EQE as a function of monochromatic wavelength recorded for PSCs based on MAPbI₃–TMTA after cross-linking. **e** The *J–V* metrics for 20 separated PSCs based on MAPbI₃–TMTA before, after cross-linking and control MAPbI₃

power output of 19.1% and 18.8% in cross-linked MAPbI₃–TMTA and control MAPbI₃ devices respectively, agreeing closely with that obtained from *J–V* curves. To understand the better performance in cross-linked MAPbI₃–TMTA devices, we further study the recombination mechanism in PSCs through estimating the ideality factor ($n$)[18]. As shown in Supplementary Fig. 7, we obtain $n$ of 1.78 and 1.50 in devices with MAPbI₃ and cross-linked MAPbI₃–TMTA by fitting the $V_{oc}$ under different light intensity. The high $n$ value indicates the severer monomolecular recombination in control MAPbI₃ devices, agreeing with the results obtained from PL and TRPL.

Figure 4d shows the external quantum efficiency (EQE) spectra of our PSCs. The cross-linked MAPbI₃–TMTA devices exhibit high EQE value (over 80%) in the visible light region (410–740 nm) and the integrated current densities closely agree with those extracted from *J–V* curves. Fig. 4e summarizes the statistical distribution of device parameters among 20 separated PSCs. The high performance is reproducible in devices with cross-linked TMTA and an average $V_{oc}$ of 1.08 ± 0.02 V, $J_{sc}$ of 22.2 ± 0.52 mA cm⁻² and FF of 77.0 ±

1.4% is obtained (Supplementary Table 1), resulting in an average efficiency of 18.43 ± 0.45%. For control MAPbI₃ devices, the average efficiency is 17.98 ± 0.63%, with $V_{oc}$ of 1.06 ± 0.03 V, $J_{sc}$ of 21.5 ± 0.89 mA cm⁻², FF of 79.2 ± 1.4%.

**Long-term stability**. Apart from efficiency, we further examine the stability of PSCs under air, thermal and even operational conditions. In PSCs, the device stability is closely related to the GBs protection[30] and ions (iodide) migration[45–48] during operation. Previous study has confirmed that GBs existence is a major reason for MAPbI₃ film decomposition in air as moisture can penetrate into the film bulk through GBs and thus accelerate MAPbI₃ decomposition. In cross-linked MAPbI₃–TMTA films, the GBs are blocked by cross-linked TMTA, suppressing moisture penetration. In addition, the contact angle of water on perovskite films with TMTA significantly increases from 52° (before cross-linking) to 73° after cross-linking due to the hydrophobic nature of TMTA (Supplementary Fig. 8), which

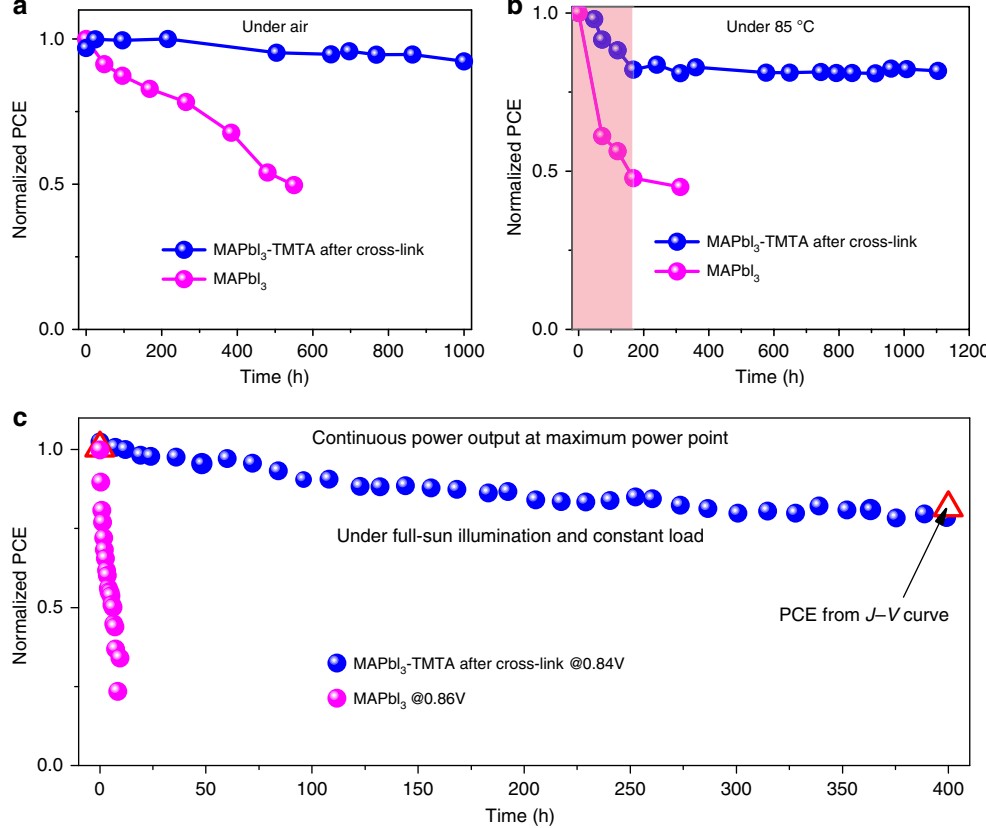

**Fig. 5** Long-term stability. **a** Air stability of non-encapsulated PSCs based on cross-linked MAPbI$_3$–TMTA and control MAPbI$_3$. The devices are kept in air (relative humidity: 45–60%) and measured regularly in glovebox filled with N$_2$. **b** Thermal stability of PSCs based on cross-linked MAPbI$_3$–TMTA and control MAPbI$_3$. The devices are kept on hotplate (85 °C) in glovebox and measured regularly. **c** Operational stability of non-encapsulated MAPbI$_3$–TMTA (after cross-linking) and control MAPbI$_3$ based PSCs. The devices are examined at maximum power point with a constant load (0.84 V, 0.86 V for MAPbI$_3$–TMTA and MAPbI$_3$ devices respectively) under continuous full-sun, AM 1.5 G illumination in glovebox. Under this condition, the continuous power output is monitored in our PSCs. The red triangle is the devices efficiency obtained from J–V curves. Note that the light source in the operational stability is Xenon lamp (Newport Oriel Sol3A solar simulator, 100 mW cm$^{-2}$) without any UV filter calibrated by a reference solar cell (Newport) and that the external load (0.84 V) and AM 1.5 G illumination is continuously applied on the PSCs except the calibration of light source

is much higher than that on control MAPbI$_3$ films (48°), thus improving the device stability in air (XRD evolution of perovskite films exposed to air shown in Supplementary Fig. 9). Under air storage (relative humidity: 45–60%), the non-encapsulated devices with cross-linked MAPbI$_3$–TMTA show substantially enhanced stability, retaining 92.3% of their highest efficiency after 1000 h (Fig. 5a, non-normalized data in Supplementary Fig. 10). While, the devices before cross-linking maintain 57.6% of their initial efficiency merely after 550 h (Supplementary Fig. 11). Similarly, only 49.7% of the initial efficiency is maintained in control MAPbI$_3$ devices under the same conditions (Fig. 5a). In addition, we have also examined the long-term stability of PSCs at high temperature (85 °C, in glove-box filled with N$_2$, Fig. 5b). The cross-linked MAPbI$_3$ devices exhibit obviously enhanced stability relative to control MAPbI$_3$ devices. After early burn-in decay over the first 170 h[16], cross-linked MAPbI$_3$–TMTA devices show almost no degradation over the next 930 h, stabilizing at over 98% of the post burn-in efficiency (over 80% of initial efficiency, non-normalized data in Supplementary Fig. 12). While MAPbI$_3$ control devices only retain 45% of their initial efficiency merely after 300 h.

Operational stability is of foremost concern in solar cells commercialization. For PSCs, the long-term operational stability has become imperative because the ions (iodide) migration in perovskite films under built-in field leads to the rapid degradation of PSCs[49–53]. As shown in Fig. 5c, despite high efficiency, the control MAPbI$_3$ devices show poor operational stability, losing over 70% of their initial efficiency merely within 10 h. Conventional strategy of using small molecule additives to passivate GBs is unable to block ions migration which may be caused by their nature of weakness and linear structure. Taking MAbI$_3$–TMTA before cross-linking for example, the devices degrade rapidly at MPP (Supplementary Fig. 13), losing 65% of the initial efficiency merely within 10 h. Such instability has been associated with the iodide migration during operation[52,53] and thus reaction at CTL/perovksite contacts[54] or even with metal electrode[50,55]. To improve the operational stability, iodide migration should be suppressed especially at the GBs[49], as GBs have much faster ions migration than the bulk, serving as an ion migration channel[56,57]. The introduction of previously reported liner polymer additive (PVP) shows limited improvement on the device degradation[30]. In devices with PVP (Supplementary Fig. 14), almost no degradation is observed at first 0.5 h (1800 s) as 99% of the initial efficiency is retained. However, with time expended, the degradation starts to accelerate and only 60% of the initial efficiency is retained after 10 h. Instead, we introduce robust and continuous network polymer at GBs through in-situ cross-linking of MAPbI$_3$–TMTA film, establishing a blocker for iodide migration and thus significantly

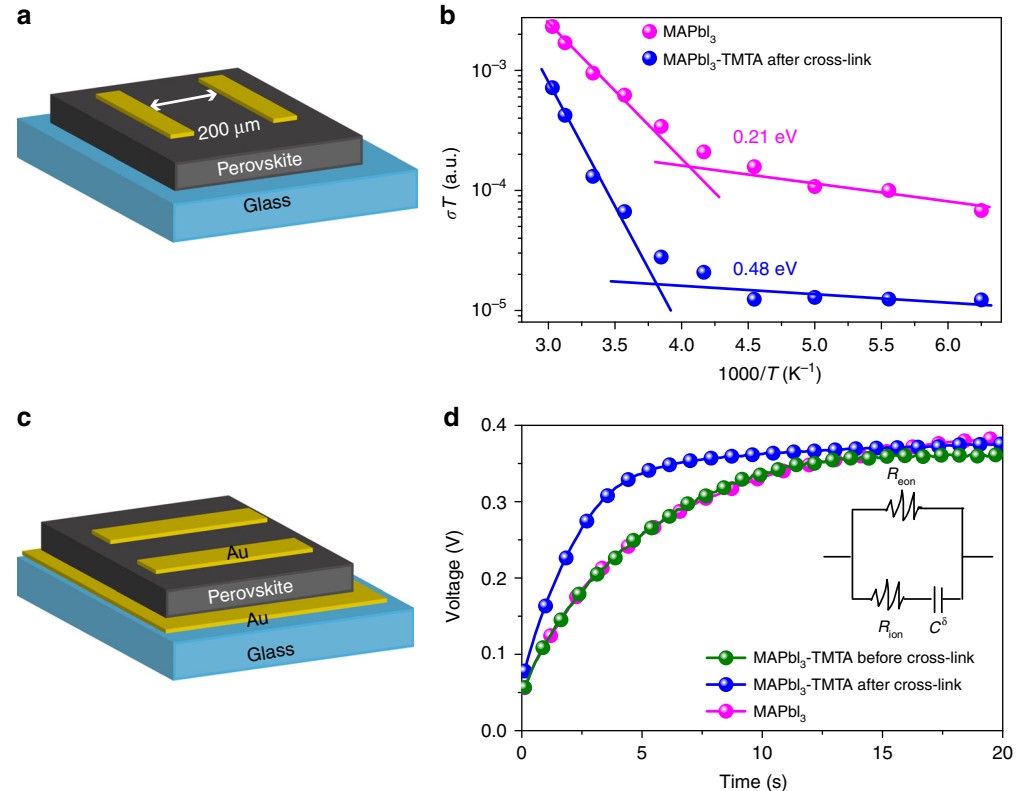

**Fig. 6** Ions migration in perovskite films. Activation energy represents how easily ions migrate. **a** Device structure used in activation energy measurement. **b** The temperature-dependent conductivity of perovskite films. Ions conductivity represents how fast ions migrate. **c** Device structure used in galvanostatic characterization. **d** Polarization curve in Au/perovskite/Au device measured in air by applying a constant current of 2 nA. The voltage response with time is recorded using Keithley 4200-SCS. Inset shows the equivalent circuit mimicking the galvanostatic characterization ($R_{eon}$ electronic resistance, $R_{ion}$ ionic resistance, $C^\delta$ chemical capacitance)

improving the operational stability of PSCs under full-sun AM 1.5 G illumination. The cross-linked MAPbI$_3$–TMTA devices retain 78.5% (81.6% from $J$–$V$ curve, red triangle) of their initial efficiency after continuous power output at MPP for 400 h with a constant load of 0.84 V (the voltage at MPP from initial $J$–$V$ curve). The current density evolution during the operational stability test and the non-normalized data are shown in Supplementary Fig. 15. The efficiency distinction between stability test and $J$–$V$ curve originates from the device degradation during operation, thus leading to the voltage at MPP variation from initial 0.84 V to 0.82 V after 400 h (Supplementary Fig. 16). We define the time of 80% degradation as T$_{80}$ to quantitatively evaluate the operational stability[16]. The cross-linked PSCs with TMTA exhibit a T$_{80}$ of 390 h, which is 590-fold higher than that of control MAPbI$_3$ devices (T$_{80}$ of 0.66 h). Note that to simulate the realistic working conditions of PSCs, full-sun illumination and constant load (0.84 V) are continuously applied on the devices during the whole operational stability test. In addition, our operational stability test is conducted using standard Xenon lamp (Newport Oriel Sol3A solar simulator, 100 mW cm$^{-2}$) without any UV-filter and the slight fluctuation in stability test should be caused by the calibration of light source. Apart from MAPbI$_3$, the in-situ cross-linking strategy is also suitable in CsFAMAPbBr$_{3-x}$I$_x$ system. As shown in Supplementary Fig. 17, the CsFAMAPbBr$_{3-x}$I$_x$ devices with TMTA after cross-linking exhibit good operational stability even in humidity air without encapsulation (relative humidity: 30–60%), retaining nearly 70% of the initial efficiency after continuous working at MPP for 100 h. In-depth investigation on the efficiency and long-term

stability of CsFAMAPbBr$_{3-x}$I$_x$ devices is beyond the scope of this work and will be the subject of future study.

**Ions migration in perovskite films**. We trace device degradation to the ions migration in perovskite films when PSCs are illuminated at MPP due to the existence of strong build-in field[50,52]. We confirm the severe ions migration in MAPbI$_3$ films through the measurement of activation energy ($E$a) and ions conductivity. $E$a of ions conduction represents how easily ions migrate and can be obtained from the dependence of conductivity on temperature in MAPbI$_3$ films[58]. We use lateral devices (Fig. 6a) in $E$a measurement to weaken the electrons conduction and thus highlight the ions conduction proportion in total current[52]. The ions migration rate in solid is determined by $E$a according to Nernst–Einstein equation: $\sigma(T) = \frac{\sigma_0}{T} \exp\left(\frac{Ea}{kT}\right)$, where $k$ is the Boltzmann constant and $\sigma_0$ is a constant. $E$a can be calculated from the slope of $\ln(\sigma T) - 1/kT$[58]. During this measurement, a small electric field of 0.1 V μm$^{-1}$ is adopted to suppress the poling effect[52]. Fig. 6b shows the conductivity of perovskite films under different temperature. The $E$a in MAPbI$_3$ film is fitted to be 0.21 eV, agreeing with previous reports[51,52]. The $E$a in MAPbI$_3$–TMTA after cross-linking is significantly increased to 0.48 eV, which is even twice larger than that in MAPbI$_3$. In addition, the threshold temperature at which ions conduction starts to dominate the total current is also impressively increased in cross-linked MAPbI$_3$–TMTA films. Ions start to migrate at 263 K in cross-linked MAPbI$_3$–TMTA films, while the threshold temperature shifts to 246 K in MAPbI$_3$ films. Both the larger

$E$a and the higher threshold temperature indicate that ions migration is much more difficult in cross-linked MAPbI$_3$–TMTA films than that in MAPbI$_3$[52].

On the other hand, ions conductivity represents how fast ions migrate in perovskite films[59]. We measure the ions conductivity of perovskite films at room temperature (298 K) using galvanostatic characterization, which is a typical method to separate ions and electrons conductivity in mixed-conductor[59]. In this measurement, a constant current of 2 nA is applied on the Au/perovskite/Au device (Fig. 6c, equivalent circuit is shown in Fig. 6d inset) and polarization curve (voltage response with time, Fig. 6d) is recorded using a semiconductor characterization system. Upon switching on the current ($i = 2$ nA), the voltage instantaneously reaches a value ($V_0$). In this stage, both the electrons and ions contribute to the electrical resistance ($V_0 = i * \frac{R^*_{eon}R_{ion}}{R_{eon}+R_{ion}}$, where $R_{eon}$ and $R_{ion}$ are electronic resistance and ionic resistance, respectively)[59]. With time increasing, ions are progressively blocked due to the formation of internal compositional gradient in perovskite films. As a result, the voltage also increases gradually until reaching a saturation value ($V$s). In this saturated region, only electrons flow and contribute to electrical resistance ($V$s $= i \times R_{eon}$)[59]. From $V$s, we can calculate the electrons conductivity ($\sigma_{eon}$) and then obtain the ions conductivity ($\sigma_{ion}$) when combining with $V_0$ (the calculated results are listed in Supplementary Table 2). The MAPbI$_3$ films exhibit high $\sigma_{ion}$ of $0.909 \times 10^{-9}$ S cm$^{-1}$, which is approximately 6-fold larger than the $\sigma_{eon}$ ($0.159 \times 10^{-9}$ S cm$^{-1}$), indicating the non-negligible ions migration in perovskite films. Cross-linking is an effective method to suppress ions migration, which has been confirmed in polymeric electrolyte system[60]. The cross-linked MAPbI$_3$–TMTA films exhibit a much lower $\sigma_{ion}$ of $0.608 \times 10^{-9}$ S cm$^{-1}$, which is decreased by 30% in comparison with that of MAPbI$_3$–TMTA before cross-linking ($0.893 \times 10^{-9}$ S cm$^{-1}$). According to the results obtained in activation energy and ions conductivity measurement, it can be concluded that ions are not only much easier to migrate but also migrate much faster in MAPbI$_3$ than cross-linked MAPbI$_3$–TMTA. Under internal field, the ions (iodide) will migrate toward and accumulate at cathode, thus accelerating the degradation of PSCs[50]. Using X-ray photoelectron spectroscopy (XPS in Supplementary Fig. 18), we confirm the accumulation of I element at interface between BCP and Cu in MAPbI$_3$ devices, which is greatly suppressed in cross-linked MAPbI$_3$–TMTA devices after operation at MPP. Basing on these findings, we conclude that the ions migration in MAPbI$_3$ films is a major reason for the devices degradation. Our in-situ cross-linked MAPbI$_3$–TMTA films can effectively suppress the ions migration through increasing the activation energy of ions migration and decreasing the ions conductivity, thus improving operational stability of PSCs.

## Discussion

In summary, we demonstrate a strategy of in-situ cross-linking organic/MAPbI$_3$ films for operationally stable PSCs through the incorporation of cross-linkable TMTA additive. TMTA can chemically bond to GBs and passivate the defects, leading to highest efficiencies of over 20%. Importantly, TMTA at GBs can be in-situ cross-linked to a robust polymer network after thermal treatment, serving as a protective layer and ions migration blocker, thus improving the thermal, water-resisting and light-resisting properties of perovskite films. The resulting PSCs exhibit 590-fold improvement in operational stability relative to control MAPbI$_3$ devices, retaining nearly 80% of the initial efficiency after continuous power output at MPP for 400 h with a constant load of 0.84 V under continuous full-sun, AM 1.5 G illumination (Xenon lamp, 100 mW cm$^{-2}$). This is the first report of operationally stable MAPbI$_3$ PSCs without any mixed-cations. In addition, the moisture or thermal (85 °C) stability is also improved, retaining over 90% of their initial or post burn-in efficiency after aging for over 1000 h.

Our work highlights the role of perovskite layer on operationally stable PSCs and proposes a unique strategy to improve the stability of perovskite layer. Using this in-situ cross-linking strategy, various cross-linked polymers with different mechanical, thermal, dielectric, water-resisting or light-resisting properties can be introduced into perovskite layer in future, thus improving the related properties of organic/perovskite films, which may be an important approach to improve the stability of perovskite films. On the other hand, the stability issues of PSCs originate not only from perovskite layer, but also from the charge transport layer (CTL). In this work, we adopt inverted device architecture with all organic CTLs (P3CT and PCBM) since our main aim is at perovskite layer. Therefore, the CTL-induced degradation is inevitable (Supplementary Fig. 18). In future studies, the operational stability of PSCs can be further improved through the combination with CTL optimization, for example, the introduction of stable inorganic CTLs, such as CuSCN[18], Ta-WO$_x$[13] and chlorine-caped TiO$_2$[19].

## Methods

**Preparation of perovskite precursor solution.** The pure TMTA is obtained from Aladdin (China). For MAPbI$_3$ precursor solution, 1.45 M PbI$_2$ (Alfa Aesar) and 1.45 M MAI are mixed together in anhydrous dimethylformamide/dimethylsulfoxide (4:1, volume ratio). For MAPbI$_3$–TMTA, TMTA is added into the precursor with a concentration of 5 mg mL$^{-1}$.

**Device fabrication.** The glass/ITO substrate (2 cm × 2 cm) is sequentially cleaned by ultrasonication in detergent, distilled water, acetone and isopropanol. The cleaned substrate is dried with N$_2$ flow and then treated in O$_2$ plasma for 2 min. The P3CT-N solution (2 mg mL$^{-1}$ in methanol) is deposited on ITO substrate in air through spin-coating at 4000 r.p.m for 60 s, and then annealed at 100 °C for 10 min. After depositing P3CT-N hole transporting layer, perovskite layer is deposited by a typical anti-solvent method in glovebox filled with N$_2$. The perovskite precursor solution is spin-coated on ITO/P3CT-N at 4800 r.p.m. for 20 s. During spin-coating, 300 μL chlorobenzene is dropped on the center of the substrate 12 s prior to the end of the program. The substrate is then annealed at 60 °C for 2 min and 80 °C for 5 min to form perovskite layer. For cross-linked MAPbI$_3$–TMTA, the perovskite film is further transfer to a pre-heated hot-plate and annealed at 140 °C 10 min to promote the cross-linking of TMTA. After cooling down to room temperature, a PCBM solution (10 mg mL$^{-1}$ in chlorobenzene) is spin-coated on perovskite layer at 2000 r.p.m. for 45 s to form electron transporting layer. Finally, the substrate is transferred into vacuum chamber; 40 nm C60, 8 nm BCP and 100 to 200 nm Cu are thermally evaporated under high vacuum ($1 \times 10^{-4}$ Pa). The active area, as defined by the overlap of Cu and ITO, is 0.06 cm$^2$.

**Characterization.** The $J$–$V$ characteristics are recorded using Keithley 2400 sourcemeter under the solar simulator (Newport Oriel Sol3A) with simulated AM 1.5 G illumination (100 mW cm$^{-2}$). The light source is a 450 W xenon lamp calibrated by a standard Si reference solar cell (Newport, 91150 V). Unless otherwise stated, The $J$–$V$ curves are all measured in glovebox at room temperature under forward scan (unless otherwise stated) from 1.2 V to −0.2 V with dwell time of 50 ms (the delay between measurement points is 50 ms). The EQE measurement is conducted in air using Newport quantum efficiency measurement system (ORIEL IQE 200TM) combined with a lock-in amplifier and 150 W xenon lamp. The light intensity at each wavelength is calibrated by one standard Si/Ge solar cell.

Fourier transform infrared spectroscopy (FTIR) samples are prepared by scraping the perovskite films off the substrate and then mixed with pre-dried KBr powder. FTIR spectra are recorded in transmittance mode using IR spectrometer instrument (Thermo, Nicolet 6700).

Top-view scanning electron microscope (SEM) images are recorded using field emission scanning electron microscope (Hitachi, S-4800) with an accelerating voltage of 8 kV. Cross-section SEM images are recorded with an accelerating voltage of 6 kV.

Transmission electron miscroscopy (TEM) images are recorded using JEOL2100, Japan. TEM samples are prepared by dropping perovskite solution (0.2 M) onto TEM grids. Then the samples are rapidly transferred into chamber linked with vacuum pump for 10–20 min. Then the samples are heated at 80 °C. For MAPbI$_3$–TMTA, the samples are further heated at 140 °C. Note that during the whole process, the TEM grids are in glass Petri dish to avoid the possible damage of grids.

STEM-EDS mapping is conducted using Tecnai F20. The sample is spin-coated on ITO substrate similar to device fabrication. Then the MAPbI3–TMTA/ITO sample is further prepared using focused ion beam (Auriga) lift-out technique.

Photoluminescence (PL) spectra were recorded with a Fluorolog-Horiba fluorometer with excitation wavelength of 370 nm. Time-resolved photoluminescence (TRPL) decays are conducted with excitation wavelength at 370 nm and emission wavelength at 770 nm.

Activation energy measurement of ions migration: The current is extracted at 150 s after the voltage is switched on. The measurement is conducted in a Lakeshore Probe Station under vacuum ($1.1 \times 10^{-4}$ Pa). The samples are placed on a copper substrate with temperature control by a heater and injected liquid He. A semiconductor characterization system (Keithley 4200-SCS) is used for the current measurement. During the measurement, we first cool the devices to 10 K for 1 h and then heat to objective temperature. Every objective temperature is stabilized for 5 min before the current record.

Ions conductivity measurement: The measurement is conducted in air by applying a constant current on Au/Perovskite/Au device (device area: 150 μm × 1000 μm). The voltage response with time is recorded using Keithley 4200-SCS.

**Stability measurement**. Dark stability is recorded by storing the non-encapsulated devices in air with controlled humidity (45–60%). Before *J*–*V* measurement, the PSCs are put into a vacuum chamber for 10–20 min to remove the moisture absorbed on surface.

Thermal stability is recorded by storing the non-encapsulated devices on a hot plate setting at 85 °C in glovebox. The *J*–*V* curves are measured after cooling the PSCs down to room temperature.

Operational stability: The *J*–*V* curves are recorded first to verify the voltage at maximum power point (MPP). Then an external bias identical to the voltage at MPP in initial *J*–*V* curves is applied on the devices. By monitoring the current density under AM 1.5 G illumination, the operational stability test is obtained through multiplying the current density by the applied bias. The AM 1.5 G illumination is achieved with a Xenon lamp (Newport Oriel Sol3A solar simulator) and periodically calibrated by a standard Si reference solar cell (Newport, 91150 V). Note that during the whole MPP tracking, the external bias and AM 1.5 G illumination is continuously applied on the PSCs except the calibration of light source. The temperature of PSCs during MPP tracking is monitored with an IR thermal gun.

## Data availability

All relevant data are available from the authors on request.

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

## Acknowledgements

This work was supported by National Natural Science Foundation of China (51773213, 61474125), China Postdoctoral Science Foundation funded project (2017M610380), Key Research Program of Frontier Sciences, CAS (QYZDB-SSW-JSC047), K. C. Wong Education Foundation (rczx0800), Zhejiang Province Science and Technology Plan (2018C01047) and National Youth Top-notch Talent Support Program.

## Author contributions

J.F. conceived the idea and supervised the whole project. X.L. designed and participated in all the experimental sections. Wenxiao Zhang and Y.-C.W. carried out the SEM, TEM and FTIR measurements. Wenjun Zhang and H.-Q.W. carried out the PL and TRPL characterization and helped analyze the results. X.L. and J.F. co-wrote the paper.

## Additional information

**Competing interests:** The authors declare no competing interests.

