## [Peer Review File · Nature Communications]

Editorial Note: Parts of this peer review file have been redacted as indicated to remove third-party material where no permission to publish could be obtained.

Reviewers' comments:

Reviewer #1 (Remarks to the Author):

I have reviewed this manuscript before while it was submitted to Nature Photonics. I think the idea of this paper is very nice, the authors have almost replied and also addressed my previous concerns, the paper could be accepted for publishing in Nat Commun. Before publication, two suggestions:

1. Please add the results about CsFAMAPbI₃-xBr in supplementary information, and discuss it in the main manuscript.
- 2) I suggest the authors to characterize the microstructure of Tmta/perovskite mixture system by high resolution transmission electron microscopy (HRTEM) to confirm the states of Tmta in perovskite layer. Similar results have been shown in recent publication doi.org/10.1016/j.chempr.2018.03.005.

Reviewer #2 (Remarks to the Author):

I want to thank the authors in addressing my comments carefully. The authors provided more data and explanation which are good. But there are still some major issues to be addressed before I can recommend its publication. The authors claimed the advantage of the 3D Tmta network on grain boundaries several times both in the manuscript and in the response letter. However, I do not see there is any clear visualization of the 3D network. The only measurement is FT-IR, but the signal is not clear and it is far from enough to proof their idea. The reaction of Tmta with other species in perovskite films can not be well excluded. The distribution of Tmta is unclear either. This needs to be seriously addressed. The authors need to show the real structural information of the crosslinked perovskite before publication. In addition, there are still some other minor issues to be addressed:

1. The authors carried out FT-IR spectra to verify the cross-linking of Tmta. But we still noted that there may be signals of CH= and CH₂= after crosslinking. The authors should show the magnified FT-IR spectra of the crosslinked samples.
2. The author claimed "the Tmta in perovskite films can chemically anchor to grain boundaries and then in-situ cross-link to a robust continuous polymer network" in the main text. This is the main difference of this work compared with previous studies, such as PEI, PEIE, etc. However, there is still not enough experimental results to support the above conclusion. The EDX elemental mapping shows Pb and O singles out of the perovskite films. Is these elements diffuse into ITO or other layers? The resolution of SEM may be not enough to probe the elemental distribution in perovskite films. EDX mapping under

high-resolution TEM or other advanced measurements are expected to demonstrate their viewpoint.

3. Tmta monomer is water soluble, which has hydrophilic functional groups. The author shows the water contact angle is approaching 73 degree, which is close to the hydrophobic region. This is surprising, because that the $-O-C=O$ group will still interact with water molecules after it is crosslinked. As the crosslinked perovskite will decompose in water, the grains should not be fully covered by Tmta at the atomic scale. And concerning that the water molecules can be adsorbed by Tmta, the reason for enhanced moisture stability should be more carefully studied and claimed.

4. The authors provides some experimental indications of suppressed ion migrations. In fact, there will be a lot of variations in these tests, such as the non-uniform coating of ETL, storage time, etc. We suggest the author provide more direct evidence of ion migration, such as the measurement of activation energy, etc.

Reviewer 1

I have reviewed this manuscript before while it was submitted to Nature Photonics. I think the idea of this paper is very nice, the authors have almost replied and also addressed my previous concerns, the paper could be accepted for publishing in Nat Commun. Before publication, two suggestions:

Response: We appreciate for the support of our manuscript.

1. Please add the results about CsFAMAPbI_{3-x}Br in supplementary information, and discuss it in the main manuscript.

Response: Thanks for the kind recommendation. In the revised version, we have added the data of CsFAMAPbBr_{3-x}I_x in supporting information as **Figure S17**. In addition, in the main text about operational stability of Tmta based perovskite solar cells (PSCs), we also add some discussion about CsFAMAPbBr_{3-x}I_x PSCs to make the work more fulfilling. In CsFAMAPbBr_{3-x}I_x with cross-linked Tmta, the initial device efficiency is 17.36% (the slightly low efficiency should be due to our imperfect fabrication process in CsFAMA system at present). The CsFAMAPbBr_{3-x}I_x devices with cross-linked Tmta show good operational stability under LED lamp (MPP tracking) even in humidity air (relative humidity: 30-60%) without encapsulation, retaining ~70% (67.6%) of the initial efficiency after 100 hours.

2. I suggest the authors to characterize the microstructure of Tmta/perovskite mixture system by high resolution transmission electron microscopy (HRTEM) to confirm the states of Tmta in perovskite layer. Similar results have been shown in recent publication doi.org/10.1016/j.chempr.2018.03.005.

Response: Thanks for the helpful recommendation. To further confirm the microstructure of MAPbI₃-Tmta, we conduct the HRTEM and the results are added in **Figure 2d-2g** and **Figure S6** (see below). Different to device fabrication where the perovskite solution can be directly spin-coated on substrate,

in HRTEM, the diluted perovskite solution is dropped on the TEM grid and transferred into vacuum chamber for 10-20min to remove the solvent. After that, the TEM grid is heated on hotplate. In low-magnification TEM image of MAPbI₃-Tmta, the MAPbI₃ grains and grain boundaries (GBs) can be seen clearly (**Figure 3d**). To confirm the nanoscale structure in MAPbI₃-Tmta, HRTEM is further conducted because cross-linked Tmta is amorphous phase while MAPbI₃ is crystalline phase (*Chem 4*, 1404-1415, (2018)). The MAPbI₃ films are fully crystalline without any obvious amorphous regions at GBs (**Figure 3e**). The MAPbI₃-Tmta films (Tmta-5mg) clearly show that amorphous walls exist among crystalline MAPbI₃ grains (**Figure 3f**), which should be ascribed to cross-linked Tmta. And the amorphous Tmta walls become much clearer in MAPbI₃-Tmta films with high Tmta concentration (Tmta-20mg in **Figure 3g**, detailed HRTEM Images see below and **Figure S6**). This result strongly confirms that cross-linked Tmta mainly exists at GBs in perovskite films.

Reviewer 2

I want to thank the authors in addressing my comments carefully. The authors provided more data and explanation which are good. But there are still some major issues to be addressed before I can recommend its publication. The authors claimed the advantage of the 3D Tmta network on grain boundaries several times both in the manuscript and in the response letter. However, I do not see there is any clear visualization of the 3D network. The only measurement is FT-IR, but the signal is not clear and it is far from enough to proof their idea. The reaction of Tmta with other species in perovskite films can not be well excluded. The distribution of Tmta is unclear either. This needs to be seriously addressed. The authors need to show the real structural information of the crosslinked perovskite before publication. In addition, there are still some other minor issues to be addressed:

Response: Thanks for the comment about our manuscript. According to comments, we have conducted more characterization about MAPbI₃-Tmta films to confirm the states of Tmta in the revised version, such as high resolution TEM, EDS mapping (scanning TEM mode), ¹H NMR, activation energy of

ions migration and ions conductivity. And we hope the results can well answer the comments and make our work more substantial.

We think there may be some misunderstanding about our manuscript. The “3D polymer network” of Tmta we say in the manuscript is in comparison with common linear polymer. Polymers can be divided into two types, linear polymer or cross-linked polymer. From the point of molecular structure, linear polymer is linear structure while cross-linked polymer is network structure. So in the field of polymer chemistry, cross-linked polymer is usually called polymer network (for example, *Science*, **334**, 965-968, (2011)). Given the comments from Reviewer, we revise the description in the main text by using “network polymer” to replace “3D polymer network” to avoid the unnecessary misunderstanding. We thanks again for the comments on this point.

In addition, more visualization measurement is conducted to confirm the polymer network of cross-linked Tmta. Just as listed below (response to Comment 1), the FTIR shows that the CH₂=CH groups of Tmta disappear in MAPbI₃-Tmta film after cross-link, indicating Tmta is fully cross-linked. Different to common linear polymer, such as PMMA, PEO, PVP, Tmta owns three cross-linkable CH₂=CH groups. When Tmta is heated, the CH₂=CH groups can be polymerized along 3 directions, forming cross-linked polymer with network structure. The major difference between cross-linked polymer and linear polymer is its insoluble and infusible nature. Cross-linked Tmta is not soluble in most solvent from strong polar (DMF, DMSO) to weak polar (Chloroform, Toluene), indicating its insoluble property (See below, the solution is heated at 80 °C for 10 hours. The insoluble solid at the bottom is cross-linked Tmta). In addition, cross-linked Tmta is still a hard solid even heated at 200 °C for 10 hours, indicating its infusible property (See below). The insoluble and infusible property of cross-linked Tmta further confirms its network structure.

High resolution TEM (HRTEM) is conducted to confirm the microstructure of MAPbI₃-Tmta (See below). In HRTEM, the crystalline phase is MAPbI₃ and the amorphous phase is cross-linked Tmta. In MAPbI₃ (Tmta-0mg), the film is fully crystalline. While in MAPbI₃-Tmta (Tmta-5mg), amorphous phase appears at the boundaries among crystalline MAPbI₃, which should be ascribed to cross-linked Tmta. And in MAPbI₃ film with high concentration Tmta (Tmta-20mg), it is much clearer that amorphous Tmta phase exists at GBs among MAPbI₃ crystalline area (detailed HRTEM images see below and **Figure S6**). The results of HRTEM strongly confirm that the cross-linked Tmta indeed exists at GBs in perovskite films.

In MAPbI₃-Tmta, there are three compositions: MAI, PbI₂ and Tmta. Tmta can coordinate with PbI₂, which has been confirmed by FTIR shift in **Figure 3b** and previous reports (*Nat. Energy* **1**, 16142, (2016); *Adv. Mater.* **30**, 1703670, (2018)). To exclude the possible reaction of Tmta with other species (MAI), nuclear magnetic resonance (¹H NMR) is conducted in pure MAI or MAI with Tmta. If MAI can react with Tmta, the chemical shift of MA⁺ should change. As shown below, proton signal of -NH₃⁺ in MAI appears at $\delta = 7.47$ ppm. When mixing MAI and Tmta together, proton signal of -NH₃⁺ still appears at $\delta = 7.47$ ppm without any shift. This result can well exclude the reaction between MAI and Tmta, supporting our description in the manuscript.

1. The authors carried out FT-IR spectra to verify the cross-linking of Tmta. But we still noted that there may be signals of CH= and CH2= after crosslinking. The authors should show the magnified FT-IR spectra of the crosslinked samples.

Response: Thanks for the comments. In FTIR spectra, the vibration of N-H bond in CH₃NH₃I (~1640 cm⁻¹, see below) may disturb the signals from Tmta, especially at the C-C stretching vibration of C=C

at 1640 cm^{-1} . That may be the reason why there “seems to be” some signals of C=C in MAPbI₃-Tmta even after cross-link.

To make the FT-IR signal clearer, in revised version, we measure the IR again using high resolution (1 nm every step, while 4 nm every step in initial version). The results are shown in **Figure 2a** and the magnified FT-IR is listed in **Figure S2**. In addition, we further confirm the characteristic peaks of Tmta according to the standard IR spectra from Spectral Database for Organic Compounds (SDBS) provided by *National Institute of Advanced Industrial Science and Technology* (AIST), Japan (see below left). (SDBS Web: <https://sdb.sdb.aist.go.jp> (National Institute of Advanced Industrial Science and Technology, accessed Aug. 9, 2018))

In Tmta, the stretching vibration of C-H in CH₂=CH (ν_{CH_2} and ν_{CH}) appears at 3043 cm^{-1} and 3110 cm^{-1} . And the C-H bending vibration in CH₂=CH (γ_{CH}) appears at 903 cm^{-1} . The vibration of C=O is very strong and appears at 1725 cm^{-1} . The vibration of C-C in CH₂=CH ($\nu_{\text{C=C}}$) appears at 1640 cm^{-1} . After cross-link, the stretching vibration ν_{CH_2} (3043 cm^{-1}), ν_{CH} (3110 cm^{-1}) and γ_{CH} (903 cm^{-1}) disappears completely (magnified IR in **Figure S2a** and **S2b**). While the vibration of C=O still exists, indicating the cross-link of Tmta through CH₂=CH after heating at $140\text{ }^\circ\text{C}$. The C-C vibration in CH₂=CH at 1640 cm^{-1} is also greatly weakened after cross-link and the weak signals around 1640 cm^{-1} in cross-linked MAPbI₃-Tmta should be from N-H vibration in MAPbI₃, not C=C (see below and **Figure S2d**).

2. The author claimed “the Tmta in perovskite films can chemically anchor to grain boundaries and then in-situ cross-link to a robust continuous polymer network” in the main text. This is the main difference of this work compared with previous studies, such as PEI, PEIE, etc. However, there is still not enough experimental results to support the above conclusion. The EDX elemental mapping shows Pb and O singles out of the perovskite films. Is these elements diffuse into ITO or other layers? The resolution of SEM may be not enough to probe the elemental distribution in perovskite films. EDX mapping under high-resolution TEM or other advanced measurements are expected to demonstrate their viewpoint.

Response: Thanks very much about the elemental mapping in our manuscript. In the revised version, we conduct cross-sectional elemental mapping of ITO/MAPbI₃-Tmta in scanning TEM mode (STEM). To in-situ characterize the elemental distribution, the cross-sectional STEM samples are prepared by spin-coating perovskite solution on ITO substrate using the same process with device fabrication. Then the sample is further processed using focused ion beam lift-out technique. The mapping area is focused on the perovskite layer, which is demarcated with red line as shown **Figure 3a** (listed below). Pb element represents MAPbI₃ and O element represents cross-linked Tmta. As shown below, O element is observed throughout the Pb-rich area, indicating the homogeneous distribution of Tmta in bulk perovskite. Combining with the results from HRTEM, it can be concluded that Tmta distributes at GBs in the whole perovskite layer.

3. Tmta monomer is water soluble, which has hydrophilic functional groups. The author shows the water contact angle is approaching 73 degree, which is close to the hydrophobic region. This is

changed as "(National Center for Biotechnology Information. PubChem Compound Database; CID=27423, <https://pubchem.ncbi.nlm.nih.gov/compound/27423> (accessed Aug. 9, 2018))

surprising, because that the $-O-C=O$ group will still interact with water molecules after it is crosslinked. As the crosslinked perovskite will decompose in water, the grains should not be fully covered by Tmta at the atomic scale. And concerning that the water molecules can be adsorbed by Tmta, the reason for enhanced moisture stability should be more carefully studied and claimed.

Response: Thanks very much for the comments about the property of Tmta. However, we think there may be some misunderstanding about Tmta. We check the chemical and physical property of Tmta in an open chemistry database provided by U.S. National Library of Medicine (<https://pubchem.ncbi.nlm.nih.gov/compound/27423#section=Top>). (National Center for Biotechnology Information. PubChem Compound Database; CID=27423, <https://pubchem.ncbi.nlm.nih.gov/compound/27423> (accessed Aug. 9, 2018) The results show that Tmta is insoluble in water (see below). To further check this property, we mix Tmta and H_2O together and find that the mixed solution will change into two layers spontaneously (**Figure S8**, see below, to make it clear to see, we add some PEO-DT:PSS to the H_2O . So the H_2O layer is blue). This result further confirms that Tmta is indeed insoluble in H_2O . When Tmta is cross-linked, the polymer is hydrophobic. This is the reason why perovskite with Tmta show large water contact angle of 73 degree.

REDACTED

As shown in HRTEM (listed above), cross-linked Tmta mainly exists at perovskite grain-boundaries (GBs) and the perovskite grains still expose outside. So if immersing perovskite with Tmta into water, the perovskite film would decompose in short time. Even for MAPbI₃ single crystal, it would also rapidly decompose if completely immersing into water. However, for perovskite film in humidity air, the situation may be different. Previous studies have proved that in polycrystalline MAPbI₃ films (solution processed MAPbI₃ films are usually polycrystalline), moisture can penetrate into the bulk of perovskite films through GBs, thus accelerating the decomposition of MAPbI₃ (*Energy. Environ. Sci.*, 10, 516-522, (2017)). This is the main reason why perovskite single crystal or films with large grain size show much improved moisture stability in comparison to that with small grain size. In our work, Tmta at GBs can block the moisture penetration into film bulk due to its hydrophobic nature, thus suppressing (not completely avoiding) the decomposition of MAPbI₃. That is the reason why cross-linked MAPbI₃ devices show much improved stability in humidity air.

At last, according to the comments, we add more descriptions about the humidity stability of PSCs in the revised version to make this point clearer.

4. The authors provides some experimental indications of suppressed ion migrations. In fact, there will be a lot of variations in these tests, such as the non-uniform coating of ETL, storage time, etc. We suggest the author provide more direct evidence of ion migration, such as the measurement of activation energy, etc

Response: Thanks for the comments and recommendation. In revised version, we further measure the activation energy (E_a) and ions conductivity of perovskite films to provide more direct evidence of ion migration. E_a of ions conduction represents how easily ions migrate and ions conductivity represents how fast ions migrate.

E_a of ions migration can be obtained from the dependence of conductivity on temperature in MAPbI₃ films (*Phys. Chem. Chem. Phys.*, **18**, 30484-30490, (2016)). We use a lateral device (**Figure 6a**, listed below) in E_a measurement to suppress the electrons conduction and thus highlight the ions conducting contribution to total current (details in main text and characterization section). The ions migration rate

in solid is determined by E_a according to Nernst-Einstein equation: $\sigma(T) = \frac{\sigma_0}{T} \exp(-\frac{E_a}{kT})$. The E_a in

MAPbI₃ film is fitted to be 0.21 eV, agreeing with previous reports (*Phys. Chem. Chem. Phys.*, **18**, 30484-30490, (2016); *Nat. Commun.*, **6**, 7497, (2015)). The E_a in MAPbI₃-Tmta after cross-link is significantly increased to 0.48 eV, which is even twice larger than that in MAPbI₃. In addition, the threshold temperature at which ion migration starts to dominate the total current is also impressively increased in cross-linked MAPbI₃-Tmta films. Ions start to migrate at 263 K in cross-linked MAPbI₃-Tmta films, while the threshold temperature shifts to 246 K in MAPbI₃ films. Both the larger E_a and the higher threshold temperature indicate that ions migration is much more difficult in cross-linked MAPbI₃-Tmta films than that in MAPbI₃.

On the other hand, we measure the ions conductivity in perovskite films using galvanostatic characterization (details in main text). Galvanostatic characterization is a standard technique to separate ionic and electronic conductivities in mixed-conductor (*Angew. Chem. Int. Ed.* **54**, 7905-7910, (2015); *Light: Sci. Appl.* **6**, e16243, (2016)). The MAPbI₃ films exhibit high ions conductivity (σ_{ion} of $0.909 \cdot 10^{-9}$ S cm⁻¹), which is approximately 6-fold larger than the electrons conductivity (σ_{eon} ($0.159 \cdot 10^{-9}$ S cm⁻¹), indicating the non-negligible ions migration in perovskite films (**Figure 6d** and **Table S2**, see below). The cross-linked MAPbI₃-Tmta films exhibit a much lower σ_{ion} of $0.608 \cdot 10^{-9}$ S cm⁻¹, which is decreased by 30% in comparison with that of MAPbI₃-Tmta before cross-link ($0.893 \cdot 10^{-9}$ S cm⁻¹).

According to the results obtained in activation energy and ions conductivity measurement, it can be concluded that ions are not only much easier to migrate but also migrate much faster in MAPbI₃ than cross-linked MAPbI₃-Tmta.

Table S2 The ions and electronic conductivity of perovskite films measured through galvanostatic characterization.			
Conductivity (10 ⁻⁹ S cm ⁻¹)	Control MAPbI ₃	MAPbI ₃ -Tmta before cross-link	MAPbI ₃ -Tmta after cross-link
σ_{ion}	0.909	0.893	0.608
σ_{eon}	0.159	0.167	0.160

σ_{ion} : ions conductivity; σ_{eon} : electronic conductivity.

REVIEWERS' COMMENTS:

Reviewer #2 (Remarks to the Author):

Thanks for the authors to take more efforts to address my concerns. I am very happy with HRTEM and ion conductivity measurement. This manuscript is now much more solid with the additional data. One last comment I have is the authors should cite the right reference for ion conductivity measurement to characterize the ion migration energy barrier. The reference added clearly is not the one that introduce this method. I donot need to review this again though.

Reviewer 2

Thanks for the authors to take more efforts to address my concerns. I am very happy with HRTEM and ion conductivity measurement. This manuscript is now much more solid with the additional data. One last comment I have is the authors should cite the right reference for ion conductivity measurement to characterize the ion migration energy barrier. The reference added clearly is not the one that introduce this method. I donot need to review this again though.

Response: According to the comments, we add another reference that introduces the method used in our work for ions migration energy measurement (Ref. 59, *Phys. Rev. Lett.* **109**, 075901, (2012)).